# The impacts of COVID-19 and its policy response on access and utilization of maternal and child health services in Tanzania: A mixed methods study

Elizabeth H. Shayo[1], Nahya Khamis Nassor[2], Leonard E. G. Mboera[3], Esther Ngadaya[4], Peter Mangesho[5], Mtumwa Bakari[1], Mark Urassa[6], Mohamed Seif[5], Clotilda Tarimo[7], Ame Masemo[2], Blandina Theofil Mmbaga[7], Natasha O'Sullivan[8], David McCoy[9], Giuliano Russo[8]*

1 National Institute for Medical Research, Dar es Salaam, Tanzania, 2 Zanzibar Health Research Institute, Zanzibar, Tanzania, 3 SACIDS Foundation for One Health, Sokoine University of Agriculture, Morogoro, Tanzania, 4 National Institute for Medical Research, Muhimbili Research Centre, Dar es Salaam, Tanzania, 5 National Institute for Medical Research, Amani Research Centre, Muheza, Tanzania, Tanzania, 6 National Institute for Medical Research, Mwanza Research Centre, Mwanza, Tanzania, 7 Kilimanjaro Clinical Research Institute, Kilimanjaro Christian Medical Centre and Kilimanjaro Christian Medical University College, Moshi, Tanzania, 8 Wolfson Institute of Population Health, Queen Mary University of London, London, United Kingdom, 9 International Institute for Global Health, The United Nations University, Kuala Lumpur, Malaysia

* g.russo@qmul.ac.uk

## Abstract

The SARS-Cov-2 virus (COVID-19) has had a global social and economic impact. Despite the growing evidence, its effects on access and delivery of maternal and child health services in low-income countries are still unclear. This cross-sectional case study was conducted in Mjini Magharibi, Chake Chake, and Ilala districts in Tanzania to help fill this gap. The study combined qualitative and quantitative data collection methods, providing an account of the evolution of the pandemic and the associated control measures in Tanzania. We drew from 34 in-depth interviews, 60 semi-structured interviews, and 14 focus group discussions with key informants, patients, and health providers, and complemented the findings with a review of pandemic reports and health facility records. We followed the Standards for Reporting Qualitative Research (SRQR) to provide an account of the findings. Our account of the pandemic shows that there was at times an inconsistent policy response in Tanzania, with diverse control measures adopted at various stages of the epidemic. There was a perception that COVID-19 services were prioritized during the epidemic at the expense of regular ones. There were reports of reorganisation of health facilities, reallocation of staff, rescheduled antenatal and postnatal clinics, and reduced time for health education and child monitoring. Scarcity of essential commodities was reported, such as vaccines, equipment, and medical supplies. Such perceptions were in part supported by the routine utilization evidence in the three districts, showing a lower uptake of antenatal, postnatal, family planning, and immunization services, as well as fewer institutional deliveries. Our findings suggest that, although the policy response in Tanzania was erratic, it was rather

**Data Availability Statement:** The anonymised interviews database has been uploaded as Supporting Information.

**Funding:** This study was financially supported by the UK Engineering and Physical Sciences Research Council COVID-19 Agile Response Fund project in the form of a grant (GCRF_NF391) awarded to EHS, DMC and GR. No additional external funding was received for this study. The funders had no role in study design, data collection and analysis, decision to publish, or preparation of the manuscript.

**Competing interests:** The authors have declared that no competing interests exist.

fear of the pandemic itself and diversion of resources to control COVID-19, that may have contributed most to lower the utilization of mother and child services. For future emergencies, it will be crucial to ensure the policy response does not weaken the population's demand for services.

# 1 Introduction

By December 2022, more than half a billion cases of COVID-19 had been recorded worldwide, and the pandemic had officially claimed the lives of 6.4 million people [1]. Although its effects have been felt across the world, the actual impact of the pandemic has varied hugely depending on context. Both the direct impact of COVID-19 on mortality and morbidity, as well as the indirect impacts of the communicable disease control measures implemented to prevent transmission, have been experienced differently from region to region, from country to country, and from community to community. The indirect effects of COVID-19 on health occur through multiple pathways [2], including through reduced supply of health services due to resource and personnel shortages caused by lockdown measures; weakened demand for care because of travel restrictions and fear of contagion; and the knock-on effects of scarce resources being diverted toward COVID-19 [3].

Documented accounts of the indirect harms of other infectious disease epidemics include the loss of children's learning from prolonged school closures [4], decreased access to basic administrative services [5], and traveling restrictions and border closures to contain transmission [6]. Significant disruption to the supply and demand of maternal and child health services was reported [7], which is something that had previously been observed also during the Ebola Virus Disease (EVD) epidemic in Guinea, Sierra Leone, Liberia, and the Democratic Republic of Congo (DRC) [8]. In the DRC, EVD was associated with additional delays in care for women experiencing pregnancy complications, leading to adverse outcomes [9]. In Sierra Leone, a decline in reproductive and child healthcare uptake among women was reported because of fear of contracting EVD in health facilities [10], while some women in DRC were reluctant to seek care from health facilities due to the fear of being transferred to Ebola treatment camps. A recent review [11] concluded that epidemic-related lockdowns carry heavy consequences for the health of women and children, and that governments should weigh the trade-offs of introducing such measures in low-income settings.

The suspension of basic curative and preventive health services is another indirect impact of epidemics, including the effort to maintain services during epidemics at a time when critical resources such as human power, funding, and medical supplies are redirected to emergency services [12]. In this respect, during the SARS epidemic in Taiwan in 2003, ambulatory care services decreased by 23.9%, while inpatient care dropped by 35.2% [13]. During the EVD outbreak in Sierra Leone, antenatal care coverage was reported to have decreased by 22%, and the coverage of family planning declined by 6%, alongside disruption to facility delivery and postnatal care [14].

Little is known about the development of the pandemic in Tanzania, although there were complaints that the government was keeping an ambiguous and inconsistent position on COVID-19, by not publishing surveillance data [15], or implementing comprehensive lockdown measures, and by taking initially a sceptical view of the newly discovered vaccines [16] as it waited for additional evidence on their safety and efficacy. In this paper, we present a descriptive analysis of the impact of the COVID-19 pandemic in Tanzania drawing from multiple sources of information. We describe what is known about the epidemic itself, as well as the policy response, and direct and indirect effects and impacts. At present, there are few

published empirical accounts of the impact of COVID-19 from Tanzania. The lessons from this experience will inform future pandemic preparedness plans for Tanzania, as well as for other low-income countries.

## 2 Methods

With a view toward constructing a holistic account of the COVID-19 pandemic in Tanzania, we employed a country case study design using quantitative and qualitative evidence from multiple sources. The study relied on primary and secondary data, including in-depth interviews of key informants, focus group discussions (FGDs), semi-structured household interviews, and documental analysis of published and grey reports. The country case study included an in-depth nested study of three purposively selected districts from three regions in Tanzania mainland and Zanzibar islands: Mjini Magharibi for Unguja, Chake Chake for Pemba, and Ilala for Dar-es-Salaam. The rural–urban criterion was considered, with Mjini Magharibi and Ilala representing urban districts and Chake Chake representing rural settings. The differences in such regions helped in captuiring the views on distinct policy responses. We set out to provide an account of the pandemic itself, and of the policy responses and impacts on mother and child healthcare (MCH) services.

### 2.1 Sources of information and data

To produce an account of the pandemic, we retrieved data on COVID-19 cases, deaths, and vaccinations from the World Health Organization (WHO) Coronavirus Dashboard, compiling data on the number of laboratory-confirmed cases and deaths and triangulating such information with the data made available by the Ministry of Health of Tanzania and district health administrations at the time.

For the country's policy response, we retrospectively and prospectively collated information on dates and details of government mandates, complementing this with information collected by the University of Oxford's COVID-19 Government Response Tracker (OxCGRT) and WHO's global health data observatory. The OxCGRT project has collected data from over 180 countries on 13 government policy responses to COVID-19, including 6 that fall within our definition of "lockdown measures": 1) workplace closures; 2) school closures; 3) restrictions on public gatherings; 4) stay-at-home requirements; 5) restrictions on movement; and 6) international travel controls. These data were triangulated and combined into a policy response timeline and validated by the research team in the field in Tanzania.

To produce an account of the impact on MCH services, we used i) qualitative data from our in-depth interviews, FGDs, and semi-structured household interviews; and ii) analysis of routine utilization data for mother and child services from facilities in the selected study sites. We used data from 10 facilities providing MCH services, of which 5 were from Unguja Mjini and 5 were from Chake Chake district, covering hospitals, health centers, and dispensaries. Data were also extracted from the District Health Information System (DHIS2).

Data included routine data on the uptake of child, maternal, and reproductive health services, as well as indicators of sexual and reproductive health such as immunization services, antenatal and postnatal attendance, health facility deliveries, deaths, caesarean sections, outpatient attendance, maternal deaths, and hospital admissions. The data collected covered the period from January 2019 to April 2021 to capture the variation of utilization before, during, and after pandemic.

### 2.2 Data collection for the interviews

**2.2.1 Recruitment of study participants.** In each district, one ward was selected for community interviews and FGDs. In each ward, stratification was done to represent wealthier,

**Table 1. Study participants by type of data collection method and location.**

| Method | Participant | Number of participants | | | | Total |
|---|---|---|---|---|---|---|
| | | National | Regional | District | Community | |
| In-depth interviews | Key informants and community influential and government leaders | 8 | 2 | 11 | 13 | 34 |
| FGDs | Healthcare workers, community members, and community leaders | – | – | 36 | 108 | 144 |
| Semi-structured interviews | Household members | – | – | – | 60 | 60 |
| Service utilisation record sets | Health facility records | – | – | 10 | – | 10 |

middle, and poor households; thus 20 households were selected to represent three categories in each ward. Households were then selected randomly based on a sampling frame designed to achieve a balanced mix of household types. A typology of households was developed for this purpose, with each household having at least one member of reproductive age. Recruitment of key informants and healthcare workers was done by researchers in collaboration with the national and district focal persons on health issues, while the community leadership was involved at community level. We recruited informants to match our initial sampling framework, and stopped recruiting extra interviewees when saturation point was reaching, as themes and categories started repeating in the interviews, and no new topic was emerging.

**2.2.2 Interviews with key informants.** A total of 34 in-depth interviews were conducted with informants, community and government leaders (Table 1). These included representatives from the Zanzibar Ministry of Health, the Ministry of Education of Tanzania, the United Nations Children's Fund (UNICEF), WHO, the Global Fund to Fight AIDS, Tuberculosis and Malaria, and non-governmental organizations (NGOs) interested in reproductive health issues, including the Tanzania Media Women's Association (TAMWA), *Chama cha Uzazi na Malezi Bora Tanzania* (UMATI), and the Zanzibar Female Lawyers Association (ZAFELA). In addition, medical officers, reproductive health coordinators, and educational officers at regional and district levels were interviewed. Further interviews were also conducted with community leaders, traditional healers, traditional birth attendants, and community health workers from the selected wards.

**2.2.3 Focus group discussions.** A total of 11 FGDs were conducted at community level in Chake Chake, Mjini Magharibi, and Ilala, with a total of 108 people participating in such groups. These included community members (separate groups of adult and adolescent, divided into male and females to capture MCH services specific users) and community leaders. In addition, three FGDs were conducted involving healthcare providers working in the reproductive health sections from different health facilities in the study districts. The FGDs collected views and experiences related to COVID-19 control policies and how they impacted the access to reproductive health services for mothers and children. Each FGD discussion lasted between 90 and 120 minutes.

Data collection guides were predesigned to capture important aspects such as child and mothers' services and how they were impacted by COVID-19, and factors that contributed to such impacts. The interviews were conducted in Kiswahili or English in a private and quiet setting, lasted between 45 and 60 minutes, and were all audio-recorded.

**2.2.4 Household interviews.** A total of 60 semi-structured household interviews were conducted where two members from each household were interviewed using the same guide. The aim of interviewing the second person was to cross-check the information provided by the first interviewee in the respective household by adding or clarifying some of the information.

## 2.3 Data analysis

Recorded interviews and FGDs were transcribed verbatim and later translated into English. One of the investigators went through the transcripts, comparing them against the audio

recording to confirm the correctness of the transcription. The second stage involved familiarization with the interviews by recording any analytical notes, thoughts, and interesting impressions related to the study's objectives. Multiple investigators went through the transcripts to identify codes using a thematic deductive analysis approach, and the transcripts were later reviewed by one senior member of the data collection team (either ES, LM, or BTM) to ensure consistency. Data were then coded following the coding framework, which included broad themes such as general availability of reproductive and child health (RCH) services, vaccination services, family planning services, antenatal and postnatal services, and outcomes. Patterns of responses, including similarities and differences, were documented. The codes formed the themes that were used in presenting the findings. The principles of the framework by Roberton et al. (2020) guided data extraction and analysis [17]. We used (a) method triangulation, (b) investigator triangulation, (c) data source triangulation to validate the findings across quantitative and qualitative data and different types of informants to enhance the internal validity of the findings.

The 21-item checklist from the Standards for Reporting Qualitative Research (SRQR) was used to inform the qualitative approach of the research and report on the interview findings [18].

## 2.4 Ethical considerations

This study received ethical clearance from Tanzania's National Institute of Medical Research (ref: NIMR/HQ/R.8a/Vol.IX/3742). Permission to conduct the study was obtained from the regional and district authorities. Participants were read upfront the study objectives, informed of their rights to withdraw, and asked authorisation to record. Then they were asked at the end of the interviews to sign their informed consent to use the information provided during the interviews. Anonymity and privacy were maintained throughout the study. We anonymised the transcripts by deleting names and references to specific actors. An alpha-numerical code comprising the location of the interview and a sequential number of the person interviewed was adopted to identify specific respondents and interviews without compromising anonymity.

## 3 Results

Prior to the pandemic, there had been a noticeable improvement in health system performance in Tanzania attributed to human resources and financial allocations. Tanzania's Demographic Health Survey indicated that the under-five mortality rate declined from 81 deaths per 1,000 live births in 2010 to 67 deaths per 1,000 live births in 2016; the use of modern contraceptives among married women increased slightly from 27% in 2010 to 32% in 2016; and almost all women (98%) who had given birth in the five years preceding the survey had received antenatal care from a skilled provider at least once for their last birth. Of those women who had given birth in the previous five years, 64% had been delivered by a skilled provider and 60% had been delivered in a health facility [19].

In the sections below, we first outline the specific measures that were taken in response to the evolution of the pandemic drawing from the documental analysis. We then present the qualitative evidence on the perceived and measured impact on reproductive and child services, triangulated with the available data on service utilization. The section concludes with the qualitative and quantitative findings regarding the demand for healthcare and perceptions of the availability of specific mother and child services.

## 3.1 The pandemic and the policy response measures

The first COVID-19 case was recorded in Tanzania on March 16, 2020 in the city of Arusha in a traveller returning from Belgium. The second case was recorded in Dar-es-Salaam on March

18, 2020, and the infection later spread to the rest of the regions. The government immediately reactivated the epidemic response team and the Emergency Operations Centre. The main interventions implemented included case detection and investigation, contact tracing, testing, case management, infection prevention and control, risk communication, and the promotion of public health measures to slow down and contain the COVID-19 outbreak. These were enforced through the guidelines published for COVID-19 case management and infection prevention control for health workers and the public in line with WHO recommendations. The public were asked to report suspected COVID-19 cases and triage was instigated in most health facilities, with mandatory temperature screening upon entry. By September 2022, a total of 39,341 confirmed cases and 845 deaths had been captured [20].

Tanzania's policy response to COVID-19 was distinguished by three distinct phases. To begin with, in March and April 2020, Tanzania undertook a set of responses that was similar and common to other countries in the region. In response to the first COVID-19 case, the Ministry of Health immediately reactivated the Rapid Response Teams at the national, regional, and district level. In addition, the Tanzanian government implemented various WHO-recommended measures and issued a collection of guidelines on topics that included maintaining provision of routine services, mandatory quarantine of travellers, and the utilization of personal protective equipment (PPE).

On March 17, 2020, all preschools and primary and secondary schools were closed, initially for one month but then extended indefinitely. On March 19, 2020, universities and other higher education facilities were also closed. From March 23, 2020, all incoming travellers from countries affected by COVID-19 were subject to a mandatory institutional quarantine for 14 days at their own cost in government-designated facilities. Citizens were encouraged to avoid travelling within and between the affected regions. but there was no home confinement order. The government decided not to implement a strict lockdown, fearing the economic cost and the disruption to provision of and access to routine health services, which in Tanzania would have compounded the large burden of infectious diseases (such as tuberculosis and HIV) [15]. However, some individuals and families voluntarily self-isolated when they had been in contact with an infected person. Call centres were established to allow citizens to report any suspected case, or to seek clarification on issues related to COVID-19 (Fig 1).

At the time of school closures, wearing of facemasks became mandatory on public transport, and all individuals were encouraged to wear facemasks whenever they were outside their homes. Public health messages regarding hand hygiene were accompanied by the provision of portable water carrying devices, with soap and sanitizers to facilitate handwashing. Bus terminals, business areas, churches, and mosques were required to have handwashing facilities.

International air travel was suspended from April 12, 2020, but Dar-es-Salaam port, which handles around 95% of all international trade, was kept open. Cross-border movement of cargo trucks was permitted, but when a truck driver tested positive at the border between

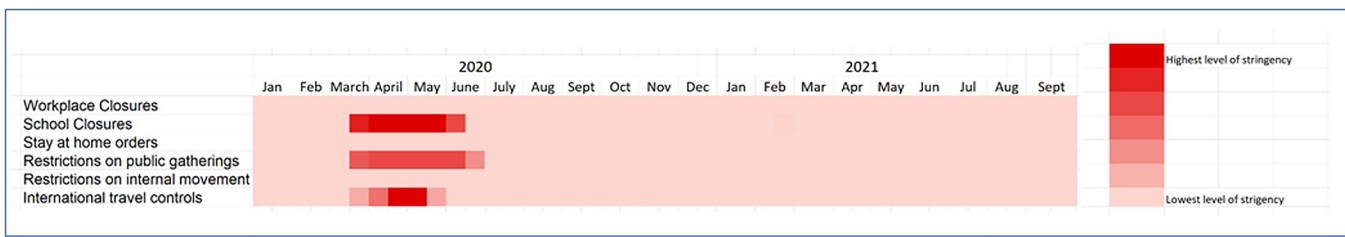

**Fig 1. Tanzania's policy response according to the Oxford stringency index (March 2020–September 2021).** Source: University of Oxford (2022) COVID-19 OxCGRT.

Tanzania and Kenya, both countries agreed to provide testing services for truck drivers as of May 7, 2020.

When the country observed a declined trend of COVID-19 cases and deaths around mid-May 2020, the government embarked on a different approach. The ban on international flights was lifted on May 18, 2020, and international flights resumed from May 27, 2020. Screening at the port of entry continued.

On May 29, 2020, the President announced all schools would reopen and that sports would resume from June 1, 2020. Universities and high schools reopened, and assessment and mitigation measures were put in place by the government to protect students and teachers. Other non-conventional approaches included the promotion of herbal and plant-based remedies as treatment for COVID-19.

## 3.2 Impact on mother and child services

**3.2.1 Suspension of clinics and services.** In response to the pandemic, some clinics and services in the three study sites were reported to have been suspended because some of the units and/or spaces were either used for management of COVID-19 patients or had inadequate human resources to maintain normal operations. For example, in Ilala district, manual vacuum aspiration during delivery was stopped in one of the facilities to allow more space to handle COVID-19 patients.

> *"You know our health facility is so small so what I did was to stop providing manual vacuum aspiration services, and used such room for handling COVID-19 patients. Sadly, to date the service has not resumed to its normal operation. If we receive clients, we refer them to district hospital." (FGD, healthcare provider, Ilala)*

Structural changes in the provision of specific RCH services made it more difficult for mothers to access them. In Mjini Magharibi, the number of health facilities providing family planning was reduced from six to three. There were also reports of fear of contracting coronavirus while accessing family planning services:

> *"Yes, people were frightened because in the hospitals there were many Corona patients, therefore, they thought going there would pose danger in contracting COVID-19 infection, hence they decided to wait!" (Household member, Ilala)*

**3.2.2 Disruption of the medical supply chain.** Key informants reported that there was disruption to procuring and distributing medical supplies due to border closures and the suspension of international travel. This caused procurement delays, leading to shortages of essential medicines and services, especially those sourced through the Central Medical Stores. As a result, supplies of medicines and laboratory reagents were limited, necessitating outsourcing medications from private health facilities:

> *"The availability of some medicines was a bit challenging. The available resources were directed to COVID-19 response, hence caused a shortage of some supplies. It reached a time that when you place an order to the Central Medical Stores, you are informed the medicines are out of stock. . . ." (Key informant interview, district level, Mjini Magharibi)*

Key informants and healthcare providers described how contraceptive medications were less available and services were subsequently reduced:

*"Corona had an effect. At first the drugs were inadequate . . . they were not imported. Even syringes were inadequate, so instead of giving family planning services every month, the schedule was changed to give* [them] *after every three months." (Health worker, FGD, Unguja)*

**3.2.3 Human resources reallocation, infections, and prioritizations.** In many health facilities, the in-charges and heads of department were relocated to manage COVID-19 isolation camps and other services, creating chaos and workforce shortages in some services. In particular, follow-up services were disrupted:

*". . . All the administration* [i.e., health facility in-charges and heads of departments] *and other workers who were supposed to make follow-up were the ones who also went for outreach and provide services to the COVID-19 isolation camps." (Key informant interview, district level, Chake Chake)*

The provision of services was also negatively affected due to staff members becoming infected with COVID-19:

*"We have ultrasound in our facility with only one expert, who got infected with COVID-19. Thus, the ultrasound services were suspended." (FGD, Healthcare provider, Mjini Magharibi)*

Care providers reported a change in planned activities as COVID-19 became the main priority. For example, antenatal clinics prioritized COVID-19 during health education sessions rather than other, common aspects of RCH.

## 3.3 Demand for healthcare and perceptions regarding availability of and access to health services

Different perspectives regarding the availability of, and access to, health services emerged between informants at various levels. While officials at the national, regional, and district level, health workers, and community FGD participants declared they had observed disruption in the provision of RCH services, the majority of household members–especially from Chake Chake rural district–did not observe such differences. Household members reported RCH as operating normally, while taking the necessary precautions against COVID-19:

*". . . Reproductive health services during COVID-19 pandemic were available as usual. . . . People were seeking services but were not allowed to stay long in the facility." (In-depth interview, household member, Chake Chake)*

Such statements were common throughout the interviews at the household level in the three districts. Key and health facility informants said that, in areas where services operated, they were fast-tracked to avoid clients staying a long time at the health facility. In some cases, extra space was created to allow for social distancing. Other changes made are described below.

**3.3.1 Access to family planning services.** The majority of study participants at all levels reported observing low uptake of family planning services because of changes in clinic schedules, fear of COVID-19 infection, and the unavailability of contraceptives. These changes interfered with preferred methods of family planning. Some women reported that they were given contraceptive pills for a longer time than usual to accommodate the increased time interval between clinics:

*". . . Services continued as usual, but we were given contraceptive pills to be used for months instead of our usual monthly frequency. . . ." (Household member, Mjini Magharibi)*

An organization dedicated to providing family planning services also observed this declining trend, as revealed in the following quote:

*"We have observed low turn-up for family planning services since when COVID-19 started. This thing has affected us so much as a result; we have a huge task of going back to the community to educate and sensitize them, so that they can continue seeking the services." (In-depth interview, NGO, Zanzibar)*

Healthcare providers confirmed having witnessed reduced family planning coverage both in in Chake Chake and Mjini Magharibi.

*"Family planning visits dropped. Usually, monthly coverage ranged between 70% and 100%. But during COVID-19, the visits dropped to less than 50% per month." (Health worker, FGD, Mjini Magharibi)*

Quantitative analysis reflecting the period between January and March in 2019, 2020, and 2021 reveals a decrease in family planning utilization in 2020 for continuing clients, while an increased trend was noted for new clients in 2020 and 2021 (Fig 2).

**3.3.2 Access to vaccination services.**   Key informants and healthcare providers reported a critical gap in providing immunization services. The timetable for vaccine administration to mothers and children was disrupted due to rescheduling clinic visits and shortages of vaccines and equipment. A participant in Ilala commented:

*"We faced a lot of challenges because it reached a time our health facility and neighbouring facilities ran out of all the vaccines. It was a big challenge even when we instructed them [clients] to attend another facility, they also missed them." (FGD, health workers, Ilala)*

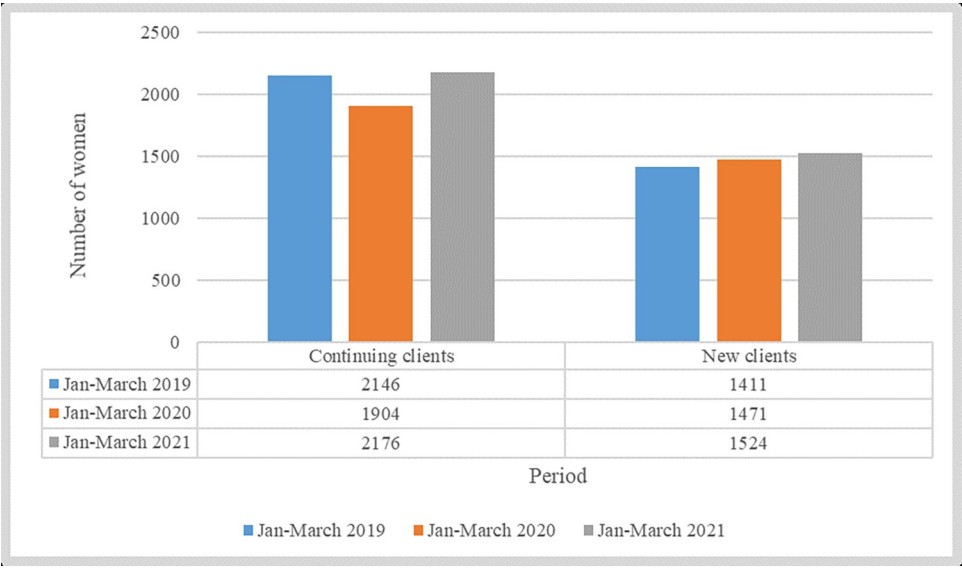

**Fig 2. Patients visiting family planning services in the selected health facilities during the first three months of subsequent years of the pandemic.** Source: Ministry of Health of Zanzibar (2022).

The unavailability of vaccines was also confirmed by community members who sought such services. Some mothers were reported to have withdrawn from child vaccination programs due to fear of catching COVID-19 infection. Key informants in Chake Chake district said community health workers were engaged to inform and educate mothers on the importance of attending antenatal clinic services, but hesitation persisted. Care providers from the same district reported reductions in immunization coverage from 100% to 28%. Surprisingly, contrary to what was reported by care providers and key informants, the majority of household members in all three districts reported the provision of child immunization services to have continued normally while observing COVID-19 preventive measures. Statements like *". . . everywhere children received immunizations and treatment services"* were common in many of the community interviews.

Despite perceived drops in immunization services from qualitative findings, quantitative data reveal similar or slightly increased trends when measured at three-month intervals (January-March 2019, 2020 and 2021). In 2021, there was increased coverage for all vaccines (except polio) in both Unguja and Pemba (Fig 3).

**3.3.3 Reduced health facility delivery.** Both key informants and healthcare providers noted a reduction in health facility deliveries, which they attributed to fear of COVID-19 among expectant mothers coupled with inadequate and unavailable services. This necessitated referral to other facilities for services such as caesarean sections, and had severe repercussions:

> *"For example, women requiring caesarean section at mid-level facility* [name hidden] *were instructed to go to the higher-level facility* [name hidden]. *This delay led to death due to complications such as postpartum haemorrhage." (FGD, healthcare provider, Ilala)*

Inadequate numbers of postdelivery beds and rooms that could maintain physical distancing interfered with the standard procedures of monitoring delivering mothers:

> *". . . What happened was that after delivery the mother was supposed to remain in the clinic for at least 24 hours. However, due to limited number of delivery rooms and beds, we did not*

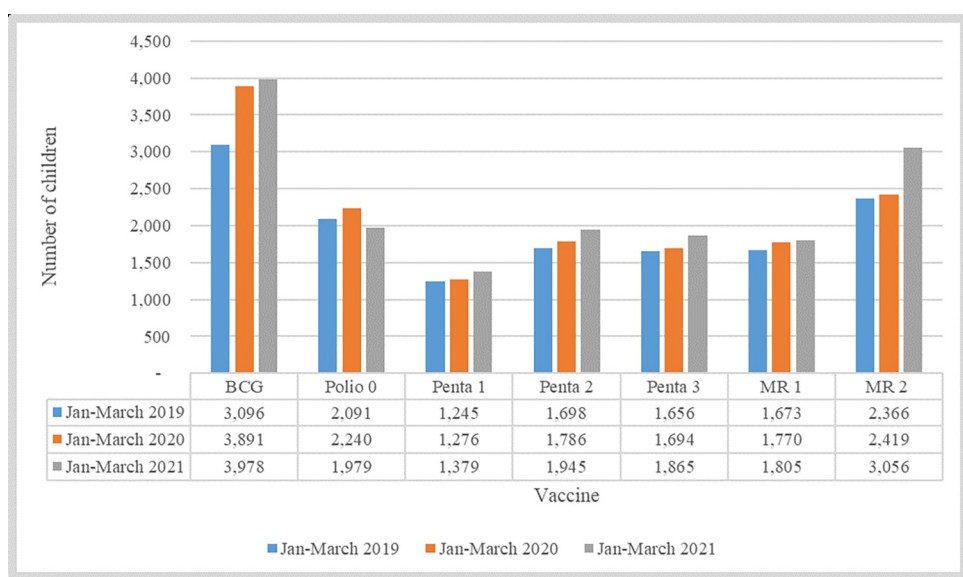

| | BCG | Polio 0 | Penta 1 | Penta 2 | Penta 3 | MR 1 | MR 2 |
|---|---|---|---|---|---|---|---|
| Jan-March 2019 | 3,096 | 2,091 | 1,245 | 1,698 | 1,656 | 1,673 | 2,366 |
| Jan-March 2020 | 3,891 | 2,240 | 1,276 | 1,786 | 1,694 | 1,770 | 2,419 |
| Jan-March 2021 | 3,978 | 1,979 | 1,379 | 1,945 | 1,865 | 1,805 | 3,056 |

**Fig 3. Number of children vaccinated from selected health facilities in Mjini Magharibi and Chake Chake, by comparable time intervals before, during, and after the pandemic.** Source: Ministry of Health of Zanzibar (2022).

*let them to stay at the clinic, so they were released after 4–6 hours." (Key informant interview, district level, Mjini Magharibi)*

A similar trend was observed by community health workers who followed up pregnant mothers and under-fives. Adult FGD members said that, even when services were available, care providers were scared to provide them to the clients. Healthcare providers in all settings confirmed observing a decline in facility deliveries and an increase in home deliveries.

However, a different scene was observed in Chake Chake, where an influential community person reported on the availability of maternity services:

*"I have been to the maternity clinic. . . . I congratulate you first and thank doctors . . . for help-ing mothers to deliver, there was no problem. . . ." (Community influential person, Chake Chake)*

Quantitative data reveal that institutional deliveries in Unguja Magharibi and Chake Chake appear to have dropped in 2020, only to rise back again to pre-pandemic levels for the same months of the year in 2021 (Fig 4).

**3.3.4 Reduced antenatal and postnatal clinic attendance.** Key informants said that visit schedules for antenatal care services were revised, with a reduction in the number of patients attending clinics per session to decongest the clinics and allow for revised staff schedules and priorities. Rescheduling visits caused delays in attending services and, in one district, partici-pants reported the services as having been unavailable for up to three months. For example, care providers were instructing pregnant women, especially those in their first trimester, to return home when there was an inadequate supply of PPE for them to use, and some were told to stay home for three months:

*". . . It is true for those who were in their 12–24 weeks of pregnancy that they were told to get back home, only to seek care when experiencing complications. Sometimes they never came back even after having some pregnant challenges because they knew there was COVID-19." (FGD, care provider, Ilala)*

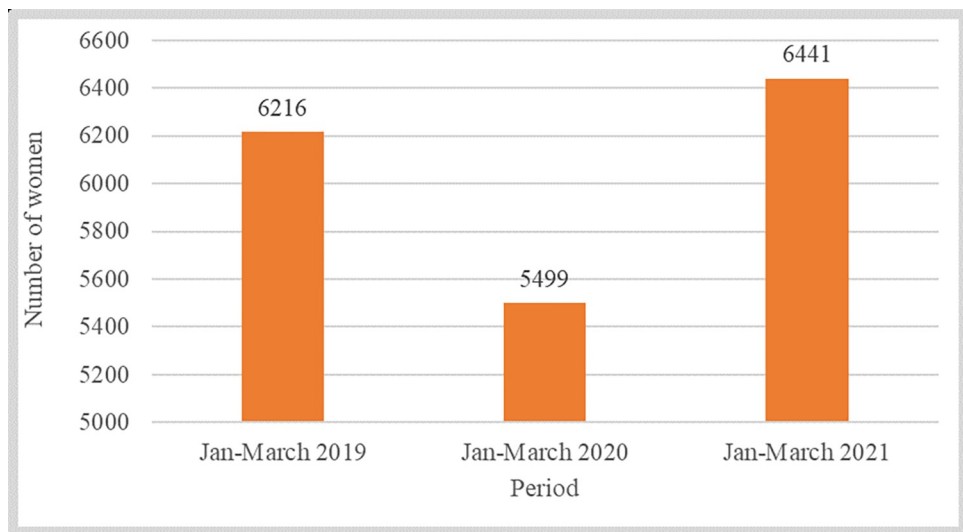

**Fig 4. Institutional deliveries recorded in the two settings in Zanzibar (three-month comparisons) over subsequent years.** Source: Ministry of Health of Zanzibar (2022).

It was reported in the FGDs with community members and healthcare providers that these changes affected foetal growth monitoring. Despite rescheduling services, patients reported fear of attending clinics and care providers reported being unwilling to deliver services to those who attended. These opinions were heard more strongly in urban than rural areas.

*"People never went to the clinic. Even care providers at the clinic were avoiding them! Pregnant mothers and children feared of Corona, thinking that they will contract it by going there . . ., and they will be isolated." (Household member, Ilala)*

This fear was attributed to the danger of getting infected with COVID-19 or of being referred to an isolation centre if COVID-19 was suspected. Community Health Workers who monitored pregnant mothers and under-fives in the community observed a similar trend, and thus monitoring of child health was not performed to the usual standard. Exactly the same reflection was mentioned in the FGDs regarding postnatal services. A participant in Chake Chake commented: *"Growth monitoring for under-fives was affected. . . . The number of pregnant women attending antenatal care clinics was affected too. . . ."* Fear of getting COVID-19 at the facility emerged as a key driver for the reduction in the demand for services.

However, when quantitative analysis was done comparing three-month intervals for three years (January to March in 2019, 2020, and 2021) in Mjini Magharibi and Chake Chake, there was an increased trend observed in all clinic intervals except the first contact at 12 weeks in 2020 (Fig 5).

**3.3.5 Reduction of services for expecting mothers.** Care providers said there were reduced growth monitoring services for pregnant women and children. Pregnancy complications such as pre-eclampsia increased because they were detected late, as testified in the following quote from a healthcare provider in Ilala:

*". . . Most of pregnant women were able to attend only two visits, hence some of them experienced pregnancy induced hypertension complications because it was not early detected.*

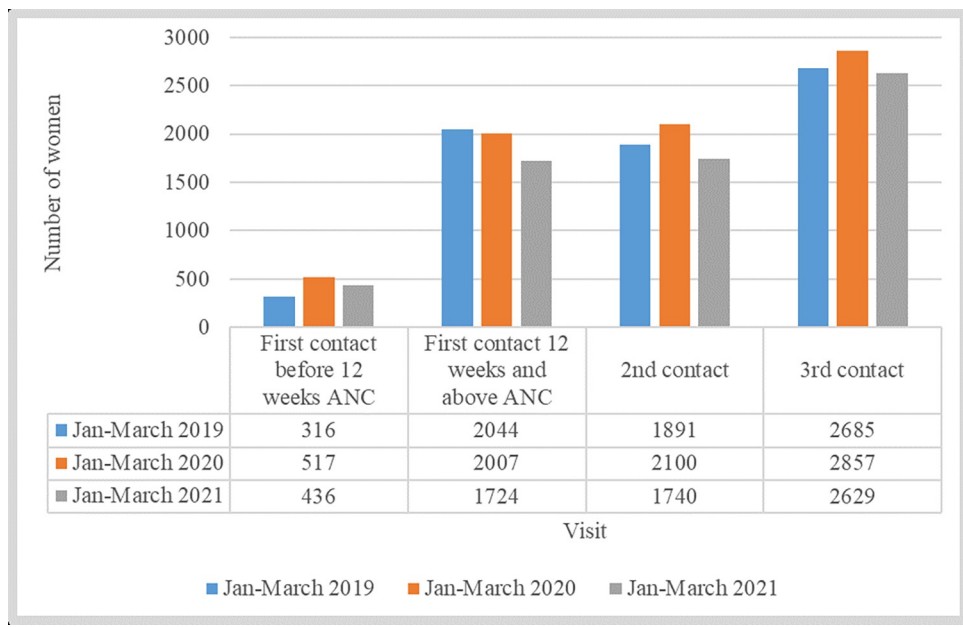

**Fig 5. Hospital visits by woman from January to March, 2019–2021, in selected health facilities in Mjini Magharibi and Chake Chake.** Source: Ministry of Health of Zanzibar (2022).

*Since they were at home, they presented at the facility when it has already reached in the bad stage. . . .*"

Higher probabilities of HIV transmission from mother to child were reported because of home deliveries and interference with prevention of mother-to-child transmission of HIV services:

"*I suspect that mothers who were HIV positive* [were] *likely to have infected their new-borns, because some of them did not share their status with anyone*, [and] *did not receive ARV.*" *(Participant, FGD, Ilala)*

## 4 Discussion

To our knowledge, this study is the first to provide an account pandemic-related events on the ground in Tanzania and to document experiences on the impact of COVID-19 and associated policies on mother and child services in that country. Our account of the pandemic shows that there was a less-than-consistent policy response in Tanzania, with diverse control measures adopted at various stages of the epidemic. Our interviewees suggested a widespread perception that COVID-19 services were prioritized during the epidemic at the expense of regular ones. There were reports of reorganised health facilities, reallocation of staff, rescheduled antenatal and postnatal clinics, and reduced time for health education and child monitoring. Such perceptions were only in part supported by the routine utilization evidence in the three districts, showing a lower uptake of antenatal, postnatal, family planning, and immunization services, as well as fewer institutional deliveries.

Our study findings need to be interpreted within Tanzania's COVID-19 context, and in part help explain its government's response measures. Although there seems to have been a substantial wave of infections in early 2021, from the limited epidemiological information available, the epidemic did not spread that extensively in Tanzania. This is consistent with the emerging information on COVID-19 excess deaths worldwide [21,22]. The reasons for this are unclear: recent data suggest that, although SARS-Cov-2 infections did happen in most African countries, these did not necessarily translate into excess deaths [23]. Unlike many other countries, Tanzania did not implement total lock down, and many of our informants appeared to support the government policy position. The policy response in Tanzania was unconventional, and, in many respects, unique. Some of the conventional control measures were put in place, and there was a clear period of time when the President and the government actively sought to deny the existence of a global pandemic. Critics argued that such a relaxed approach put the health system at risk [24], while supporters defend that the policy response was proportionate to the situation in the field [15].

Nonetheless, our study has shown that there were indirect harms associated with the pandemic in Tanzania. Fear of contagion and the diversion of resources from regular services appear to have caused the most damage, with a temporary decrease in the utilization of mother and child services. There was a reduction in both the demand and supply sides of MCH services for several reasons, including fear of contracting COVID-19; inadequate supplies of contraceptives, vaccines, and equipment; increased time intervals for antenatal and new-born clinic schedules; shortages of healthcare personnel; and health facility prioritization of the care of COVID-19 patients. The pandemic caused distress, panic, worry, and fear among healthcare providers, patients, and the community. There were government directives aimed at prioritizing measures for infection control alongside attempts to ensure basic service provision but, when healthcare systems were already operating at the limits of their capacity (as in the case of

Tanzania), the additional demands on the system resulted in panic and suboptimal performance of health workers. Such consequences were also observed in India [25]. Crucially, the result of these effects has been associated with increasing maternal mortality ratio in different contexts [26].

Reductions in immunization coverage and family planning services were attributed by our informants to the shortage of supplies, which was in turn due to restrictions in international travel that disrupted the procurement system. The knock-on effects of missed childhood vaccinations are wide and have immediate and longer-term consequences for the health of the child and community at large. A recent systematic review also noted a reduction in immunization coverage during the pandemic [27]. Fear of exposure to COVID-19 at healthcare facilities, restrictions on movement, the shortage of healthcare workers, and the reprioritization of resources from RCH to COVID-19 has been associated with low immunization coverage in multiple contexts [28] and with declines in institutional delivery rates in Bangladesh and Ethiopia [29].

Our study also noted some reduction in the number of visits to antenatal and postnatal clinics due to clinic efforts to reduce congestion in administrative, waiting, and treatment areas. However, fear of contracting COVID-19, or of being diagnosed and taken to an isolation centre, were also important factors in the reduction of antenatal and postnatal clinic attendance, which was also witnessed in Kenya during the pandemic [30]. Interestingly, while the qualitative findings indicate a reduction in MCH utilization, the quantitative data do not support this. However, this should be interpreted with caution, as much of the data was of poor quality and incomplete, an ongoing issue in many African settings prior to the pandemic.

The reduction in MCH service provision and uptake due to the pandemic observed in our study may negatively impact achievement of the SDGs in Tanzania. Reductions in antenatal and postnatal care and family planning services led to an increase in the late detection of pregnancy-related complications and of unwanted pregnancies, while facility-based deliveries reduced in number. Similar experiences have been reported in Pakistan [31]. Moreover, an increased rate of unwanted pregnancies due to decreased use of family planning services and fewer regular clinic visits, alongside an increase in working from home, have been reported from Lebanon [32]. Furthermore, reductions in MCH services leading to increased pregnancy-related complications have been associated with increased morbidity and mortality rates [25]. Our findings indicate that the pandemic may have halted, or even reversed, some of Tanzania's progress in achieving the targets set out within SDG 3, and highlight the necessity for urgent rethinking of the delivery of health services during pandemics and other global health emergencies.

We recognize our findings are affected by a few limitations. First of all, health services utilization data in Tanzania are notoriously incomplete and at times unreliable [33], and they deteriorated considerably during the pandemic. As a result, our ability to triangulate and validate the findings from the interviews was often constrained. Second, because of the difficult political and policy context in Tanzania during the pandemic [24], some study participants may not have felt able to speak openly about their experiences, despite being guaranteed confidentiality. Finally, the locations, health facilities, and wards where we conducted our study and recruited participants for the interviews may not have been entirely representative of the whole country; our quantitative data sources were also skewed toward the selected health facilities. As a result, the generalizability of our findings to other national and regional contexts may be limited.

## 5 Conclusion

Although evidence is accumulating on COVID-19 and on the effects of the pandemic, it is not yet clear how populations and health services have been affected, particularly in low-income

settings. This study explores the impact of the COVID-19 pandemic on the delivery and uptake of maternal and child health services in Tanzania. We collected quantitative and qualitative evidence from three districts to investigate the perceptions of different stakeholders around the pandemic, as well as associated policy responses and the uptake of services. We used the SRQR checklist to provide an account of the findings to analyse, triangulate, and report our qualitative findings.

We found that, although the policy response in Tanzania was at time inconsistent, with diverse control measures adopted at various stages of the epidemic, our interviewees suggested there was a widespread perception that COVID-19 services were prioritized during the epidemic at the expense of regular ones. Such perceptions were only in part supported by the routine utilization evidence in the three districts, which showed a lower uptake of antenatal, postnatal, family planning, and immunization services, as well as fewer institutional deliveries.

Although the extent to which low-income countries have been impacted by the pandemic is not clear, or what the optimal policy response should have been, our study highlights the importance of monitoring effects on the demand for healthcare services in future epidemics, particularly for vulnerable populations.

## Supporting information

**S1 File. Interviews database.**
(XLSX)

**S2 File. Service delivery utilization data by facilities.**
(XLSX)

## Acknowledgments

We thank Gloria Angolo, Zenais Kiwale, Doris Mbata and Athanasia Joseph for supporting the data collection. All the authors are particularly indebted to Dr. Leonard Mboera, who passed away before the completion of this study. Finally, we express our gratitude to all who agreed to participate in our interviews.

## Author Contributions

**Conceptualization:** Elizabeth H. Shayo, Leonard E. G. Mboera, Peter Mangesho, Mark Urassa, Blandina Theofil Mmbaga, David McCoy, Giuliano Russo.

**Data curation:** Elizabeth H. Shayo, Nahya Khamis Nassor, Esther Ngadaya, Mtumwa Bakari, Mark Urassa, Mohamed Seif, Ame Masemo, Blandina Theofil Mmbaga.

**Formal analysis:** Elizabeth H. Shayo, Nahya Khamis Nassor, Esther Ngadaya, Peter Mangesho, Mtumwa Bakari, Clotilda Tarimo, Natasha O'Sullivan.

**Funding acquisition:** Blandina Theofil Mmbaga, David McCoy, Giuliano Russo.

**Investigation:** Esther Ngadaya, Peter Mangesho, Clotilda Tarimo.

**Methodology:** Elizabeth H. Shayo, David McCoy.

**Project administration:** Elizabeth H. Shayo, Mtumwa Bakari, Mohamed Seif, David McCoy, Giuliano Russo.

**Resources:** Elizabeth H. Shayo, Blandina Theofil Mmbaga, Natasha O'Sullivan.

**Supervision:** Blandina Theofil Mmbaga, David McCoy.

**Validation:** Elizabeth H. Shayo, Leonard E. G. Mboera, Ame Masemo.

**Writing – original draft:** Elizabeth H. Shayo, Leonard E. G. Mboera, Ame Masemo, Natasha O'Sullivan, David McCoy, Giuliano Russo.

**Writing – review & editing:** Elizabeth H. Shayo, Nahya Khamis Nassor, Leonard E. G. Mboera, Esther Ngadaya, Peter Mangesho, Mark Urassa, Mohamed Seif, Clotilda Tarimo, Ame Masemo, Blandina Theofil Mmbaga, Natasha O'Sullivan, David McCoy, Giuliano Russo.

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
