## [Decision Letter · Decision Letter 0]

10 Feb 2023

PGPH-D-23-00027

The impacts of COVID-19 and its policy response on access and utilization of maternal and child health services in Tanzania: a mixed methods study

Dear Dr. Russo,

Thank you for submitting your manuscript to PLOS Global Public Health. After careful consideration, we feel that it has merit but does not fully meet PLOS Global Public Health’s publication criteria as it currently stands. Therefore, we invite you to submit a revised version of the manuscript that addresses the points raised during the review process.

We look forward to receiving your revised manuscript.

Kind regards,

Sanjana Ravi, PhD, MPH

Academic Editor

Journal Requirements:

1. Please send a completed 'Competing Interests' statement, including any COIs declared by your co-authors. If you have no competing interests to declare, please state "The authors have declared that no competing interests exist". Otherwise please declare all competing interests beginning with the statement "I have read the journal's policy and the authors of this manuscript have the following competing interests:"

3. Please provide separate figure files in .tif or .eps format only and remove any figures embedded in your manuscript file. Please also ensure that all files are under our size limit of 10MB.

4. We noticed that you used "unpublished" in the manuscript. We do not allow these references, as the PLOS data access policy requires that all data be either published with the manuscript or made available in a publicly accessible database. Please amend the supplementary material to include the referenced data or remove the references.

5. In the online submission form, you indicated that "The anonymised interviews transcripts will be deposited on Flowrepository, and will be available upon request". All PLOS journals now require all data underlying the findings described in their manuscript to be freely available to other researchers, either 1. In a public repository, 2. Within the manuscript itself, or 3. Uploaded as supplementary information.

Additional Editor Comments (if provided):

Reviewers' comments:

Reviewer's Responses to Questions

**Comments to the Author**

1. Does this manuscript meet PLOS Global Public Health’s publication criteria? Is the manuscript technically sound, and do the data support the conclusions? The manuscript must describe methodologically and ethically rigorous research with conclusions that are appropriately drawn based on the data presented.

Reviewer #1: Yes

Reviewer #2: Yes

2. Has the statistical analysis been performed appropriately and rigorously?

Reviewer #1: N/A

Reviewer #2: N/A

3. Have the authors made all data underlying the findings in their manuscript fully available (please refer to the Data Availability Statement at the start of the manuscript PDF file)?

Reviewer #1: Yes

Reviewer #2: No

4. Is the manuscript presented in an intelligible fashion and written in standard English?

Reviewer #1: Yes

Reviewer #2: Yes

5. Review Comments to the Author

Reviewer #1: Dear Authors,

Thank you for the opportunity to review your manuscript. Please find some suggestions below to improve the quality of the material in your work

Abstract

Lines 49-50: You mention that you follow standards in reporting qualitative research - what standards did you follow? Just report the specific standards that was used directly.

Lines 60-63: Despite policy response suggests it was not the policy response but fear and diversion of resources and your suggestion for the future is on policy response! This doesn’t seem logical. If fear and disruption caused the issues that you discuss, then something needs to be done about that don’t you think?

Methodology

151- There is no random sampling in qualitative research. What sampling frame was used please provide details?

Table 1 not clear

Your description of the details of data collection should match the table – please modify the table to ensure this

Sampling

Lines 137-138: Why did you choose these specific locations (Unguja Mijni and Chake Chake districts?). Can you please give some details to explain your choice of the districts

Lines 156-…: How did you go about the numbers that you report- did you observe for saturation? Please discuss this and provide details about how you detected saturation of data and how this guided your sample size

Data analysis

Lines 191-193: If multiple investigators went through the transcript for coding, them how did you achieve inter-coder reliability? Was this double blinded coding or otherwise? How did you triangulate. What sort of triangulation approach did you use?

Ethics

Lines 207-208: It is not clear to me whether you followed a two-stage consent process with an initial oral consent followed by obtaining written consent? If this is the case then you need to make it clear if the written consent was about after data collection and when exactly.

Line 209: How did you maintain confidentiality anonymity and privacy? In the methodology section of a peer reviewed publication, it is important that the readers know how you did something (that you state you did).

Results

Lines 461-473: 3.3.5 Your title says increased complications and unwanted pregnancies.

But there is no data discussed on unwanted pregnancies. Change sub-heading to reflect results presented only.

Discussion

You need to define what is linear and non-linear policy responses in the introduction so authors know the difference.

General observation

Requires a lot of work. I would urge the authors to focus the discussion on what is relevant to your paper. Currently, I am afraid the discussion has a lot of generic material on the topic

Shorten the initial part of the discussion. If you want to set the context globally and nationally in detail, you can do that in the Introduction. In the discussion just briefly introduce any global contexts that are relevant to the topic of the paper and discuss the relevance of each of your findings within the context of the body of relevant literature.

Reviewer #2: Thank you for the opportunity to review this manuscript. It is well written and I have a few minor comments

- Will the health facility records be available in the data repository?

Methods:

- Data collection, health facility records are included under qualitative methods. It seems that these records are quantitative in nature. it would be helpful to total the number of IDIs, FGDs, and SSIs conducted. In table 1, consider adding a column to total each of the different types of information. The data sources are skewed towrard district and community level data, which affects the generalizability of the data.

Results:

- this was primarily a qualitative study. Where are the quantitative results?

Discussion/Conclusion:

- it seems there are conclusions drawn without the backing of data. For example, a "non-linear policy response," "lower uptake of antenatal, postnatal, family planning, immunization services, fewer institutional deliveries."

6. PLOS authors have the option to publish the peer review history of their article (what does this mean?). If published, this will include your full peer review and any attached files.

**Do you want your identity to be public for this peer review?** For information about this choice, including consent withdrawal, please see our Privacy Policy.

Reviewer #1: **Yes: **Sunil George

Reviewer #2: No

---

## [Editor Report · Decision Letter 1]

31 Mar 2023

PGPH-D-23-00027R1

The impacts of COVID-19 and its policy response on access and utilization of maternal and child health services in Tanzania: a mixed methods study

Dear Dr. Russo,

Thank you for submitting your manuscript to PLOS Global Public Health. After careful consideration, we feel that it has merit but does not fully meet PLOS Global Public Health’s publication criteria as it currently stands. Therefore, we invite you to submit a revised version of the manuscript that addresses the points raised during the review process.

We appreciate the revisions made in response to the prior round of peer review, and feel that the revised draft is much stronger and more cohesive as a result. However, the terms "linear policymaking" and "nonlinear policymaking" remain unclear. Please either define these terms explicitly or remove them from the revised draft altogether.

We look forward to receiving your revised manuscript.

Kind regards,

Sanjana Ravi, PhD, MPH

Academic Editor

Journal Requirements:

1. We have noticed that you have a list of Supporting Information legends in your manuscript. However, there are no corresponding files uploaded to the submission. Please upload them as separate files with the item type 'Supporting Information'.
---

## [Editor Report · Decision Letter 2]

5 Apr 2023

The impacts of COVID-19 and its policy response on access and utilization of maternal and child health services in Tanzania: a mixed methods study

PGPH-D-23-00027R2

Dear Dr Russo,

We are pleased to inform you that your manuscript 'The impacts of COVID-19 and its policy response on access and utilization of maternal and child health services in Tanzania: a mixed methods study' has been provisionally accepted for publication in PLOS Global Public Health.

Best regards,

Sanjana Ravi, PhD, MPH

Academic Editor